# Research on Extraction, Structure Characterization and Immunostimulatory Activity of Cell Wall Polysaccharides from *Sparassis latifolia*

**DOI:** 10.3390/polym14030549

**Published:** 2022-01-28

**Authors:** Jing Liu, Xuemeng Zhang, Jingsong Zhang, Mengqiu Yan, Deshun Li, Shuai Zhou, Jie Feng, Yanfang Liu

**Affiliations:** 1Institute of Edible Fungi, Shanghai Academy of Agricultural Sciences, Key Laboratory of Edible Fungi Resources and Utilization (South) Ministry of Agriculture, National Engineering Research Center of Edible Fungi, Shanghai 201403, China; lj15735649029@163.com (J.L.); xmz001122@163.com (X.Z.); syja16@saas.sh.cn (J.Z.); yanmengqiu@saas.sh.cn (M.Y.); lds18764886910@163.com (D.L.); zhoushuai@saas.sh.cn (S.Z.); sytufengjie@163.com (J.F.); 2College of Food Science & Engineering, Shanghai Ocean University, Shanghai 201306, China

**Keywords:** *Sparassis latifolia*, cell wall polysaccharides, structure, immunostimulatory activity

## Abstract

The cell wall polysaccharides were extracted from *Sparassis latifolia* fruit bodies by acid–alkali and superfine-grinding assisted methods, and the chemical characterization and in vitro immunity activities of these polysaccharide fractions were studied and compared. Results showed that superfine-grinding assisted extraction exhibited the highest yield of polysaccharides (SP, 20.80%) and low β-glucan content (19.35%) compared with alkaline extracts. The results revealed that the 20% ethanol precipitated fraction (20E) from SP was mainly composed of β-(1→3)-glucan and α-(1→4)-glucan. With the increase of ethanol precipitation, the fractions (30E, 40E, 50E) were identified as α-(1→4)-glucan with different molecular weights and conformations. Cell wall polysaccharides extracted through NaOH (NSP) and KOH (KSP) extraction had similar yields with 8.90% and 8.83%, respectively. Structural analysis indicated that the purified fraction from KSP (KSP-30E) was a β-(1→3)-glucan backbone branched with β-(1→6)-Glc*p*, while the purified fraction from NSP (NSP-30E) mainly contained β-(1→3)-glucan with a small number of α-linked-Glc*p*. The two fractions both exhibited rigid chain conformation in aqueous solutions. All polysaccharide fractions exerted the activity of activating Dectin-1 receptor in vitro, and the KSP-30E mainly identified as β-(1→3)-glucan with the terminal group via 1→6-linkage attached at every third residue exhibited a stronger enhancing effect than other fractions. Results suggested that KOH extraction could be efficient for the preparation of bioactive β-(1→3, 1→6)-glucan as a food ingredient.

## 1. Introduction

The fungi of the genus *Sparassis* are edible and medicinal mushrooms and widely distributed in Europe, the United States and East Asia. Ten species of *Sparassis* were recognized based on morphological and molecular data [1], and among these species, *Sparassis latifolia* was a widely cultivated species in China. Many reports revealed that edible fungi of the *Sparassis* genus were rich in protein, vitamins, polysaccharides, etc. [2,3,4]. Polysaccharides as the main active macromolecules exhibited a wide range of biological activities including immunomodulation [5], antitumor effects [6] and antioxidation [7].

It is well-known that the biological activity of polysaccharides is closely related to their structure [8,9,10], thus the structural analysis of polysaccharides is the basis for the study of their biological activity. The fungi of *Sparassis* genus contained high levels of β-glucan, which was determined to reach around 40% [11], and the primary structure of β-glucan was β-(1→3)-glucan with β-(1→6)-linked glucose as branches [12]. Kim et al. found that β-glucan isolated from the *Sparassis* genus exhibited immune-mediated antitumor activity mainly by activating DCs and macrophages [13]. Recently, some studies on intracellular polysaccharides from *Sparassis latifolia* showed that polysaccharides obtained by water extraction exhibited strong antioxidant, immunological and antibacterial activities [14,15,16]. In addition, further studies indicated that polysaccharides isolated from the water extracts of *Sparassis latifolia* had a main chain structure of α-(1→4)-glucan with (1→6)-branches [17], and appeared as a spherical structure of varying size under scanning electron microscopy [15]. However, the studies on the structures of cell wall polysaccharides from *Sparassis latifolia* were seldom reported. According to structural characteristics, the fungal cell wall can be divided into three layers. The inner layer contains β-glucan that exhibit multiple bioactivities [18], and the middle and outer layers are mainly composed of proteins and mannose, respectively. These layers were difficult to extract with water, so there were few studies on cell wall polysaccharides. Many extraction techniques were used to release cell wall polysaccharides such as acid extraction [19], alkali extraction [20], superfine grinding extraction [21], ultrasound-assisted extraction [22] and enzyme-assisted extraction [23]. Previous reports showed that acid-alkali and superfine grinding extraction can significantly improve the yields of polysaccharides [24,25], thus they were widely used for the extraction of cell wall polysaccharides. However, few studies compared the differences in structure and activity of the cell wall polysaccharides extracted by these two methods. Therefore, in this study, the alkali–acid–alkali sequential extraction and superfine grinding-assisted extraction were used to obtain cell wall polysaccharides from *Sparassis latifolia* fruit bodies. Meanwhile, the molecular weight distributions, monosaccharide compositions, glycosidic linkage of different fractions were compared. Furthermore, the immunomodulatory activities in vitro of cell wall polysaccharides obtained by different extraction methods were investigated. The purpose of this work is to study the effect of extraction methods on the structure and bioactivities of cell wall polysaccharides, which will provide a reference for further analysis of cell wall polysaccharides from *Sparassis latifolia.*

## 2. Materials and Methods

### 2.1. Materials and Chemicals

Fruit bodies of *Sparassis latifolia* were purchased from Fujian Tianyi Industry Co., Ltd. (Ningde city, Fujian Province, China). Monosaccharide standards including fucose, galactose, glucose, arabinose, rhamnose, mannose, glucosamine, xylose, galacturonic acid and glucuronic acid were obtained from SigmaAldrich (St. Louis, MO, USA). Scleroglucan were purchased from Seebio Biotech Co., Ltd. (Shanghai, China). Aniline blue was purchased from Shanghai Sinopharm Chemical Reagent Co., Ltd, (Shanghai, China). HEK-Blue^TM^ hDectin-1b cells and a HEK-Blue^TM^ Detection kit were bought from InvivoGen (Toulouse, France). All other reagents were analytical grade and produced in China.

### 2.2. Extraction and Isolation of Cell Wall Polysaccharides

The 500 g chopped fruiting bodies were extracted twice with 10× volumes of distilled water for 1 h at 100 °C, and the residues were collected to be oven dried at 60 °C. The water-extracted dried residues were obtained by different methods to yield the cell wall polysaccharides (as shown in Figure 1), and the detailed procedures were as follows.

#### 2.2.1. Alkali-Acid-Alkali Sequential Extraction

The polysaccharides were isolated according to previous research with modifications [26]. The residues were extracted with 500 mL 1 mol/L sodium hydroxide (NaOH) at 60 °C for 30 min under sufficient stirring. After filtering, the residue was washed twice with distilled water, and the filtrate was neutralized with 6 mol/L hydrochloric acid (HCl). Then the supernatant was concentrated, dialyzed and freeze-dried to obtain the NaOH extract (NSP). The filter cake was further extracted with 500 mL 0.5 mol/L hydrochloric acid at 100 °C for 1 h in the water bath, and the suspension was filtered and washed with distilled water again. The collected supernatant was concentrated and dialyzed after being neutralized with 1 mol/L NaOH, then freeze-dried to obtain the HCl extract coded as HSP. Finally, the residues were extracted with another 500 mL of 1 mol/L potassium hydroxide (KOH) for 30 min at 60 °C and washed with distilled water, followed again by centrifugation, concentration and dialysis, and the obtained fraction was called KSP.

#### 2.2.2. Superfine Grinding Assisted Extraction

The dried residues (250 g) were milled once using a superfine grinder (BFM100B, Billion, New Taipei City, China) to prepare superfine grinding samples, and the particles passed through a 200-mesh sieve were selected for extraction. The powder was extracted twice with water at a solid to solvent ratio of 1:10 (1 h each time) at 100 °C, and the extracts were concentrated and dried to obtain SP.

#### 2.2.3. Isolation of Cell Wall Polysaccharide

For alkali and acid extracts, the yield, polysaccharide content, β-glucan content and immune activity of each fraction were comprehensively considered, and the NSP and KSP were selected for separation and purification by ethanol precipitation. The samples of NSP and KSP were suspended in 0.1 mol/L NaOH and neutralized with 5.8 mol/L CH_3_COOH at a concentration of 2.5 mg/mL. Subsequently, the ethanol was added at the concentration of 30% for purification of polysaccharide fractions, and the precipitates named as NSP-30E, KSP-30E, respectively. The SP was dissolved in distilled water, then the ethanol was added to the solution until the final concentration reached 20% (*v*/*v*) for the precipitate, and then incubated at 4 °C for 6 h. The precipitate was collected by centrifugation (8000 g for 15 min) and dissolved in water, then the solution was freeze-dried and called 20E. Finally, to precipitate the supernatant, ethanol was added at the final concentration of 30%, 40% and 50% (*v*/*v*), sequentially. The precipitates were collected and then dried (named as 30E, 40E, 50E).

### 2.3. The Total Sugar and β-Glucan Content Determination

The total sugar content of each fraction was determined by the phenol–sulfuric acid method with glucose as standard [27]. Aniline blue fluorescence assay was applied to determine the β-glucan content with Scleroglucan as standard [28].

### 2.4. Molecular Weight Distribution Analysis

Molecular weights were determined by using high performance size exclusion chromatography (HPSEC) equipped with an eight-angle laser light scattering detector (MALLS, Wyatt Technology Co., Santa Barbara, CA, USA) and refractive index detector (RI, Waters, Milford, MA, USA) [29]. The separations were carried on two SEC columns of TSK GEL G4000 PW_XL_ and G6000 PW_XL_ (7.8 mm × 300 mm, Tosoh, Tokyo, Japan). The mobile phase was 0.15 M NaNO_3_, 0.05 M NaH_2_PO_4_ and 0.02% NaN_3_ (pH 7.0), and the flow rate and column temperature were set at 0.5 mL/min and 30 °C, respectively. The *dn*/*dc* value was set to 0.146 mL/g for polysaccharide polymers, and the ASTRA software (Version 6.1.1, Wyatt Technology Co., Santa Barbara, CA, USA) was used to analyze the data. All samples (5 mg) were dissolved in 1 mL of phosphate buffer solution and centrifuged at 8000 g for 30 min, and then the solution was filtered through a 0.45 μm membrane for analysis.

### 2.5. Monosaccharide Composition Analysis

The sample (2 mg) was first hydrolyzed with 3 mL 2 M trifluoroacetic acid (TFA) at 110 °C for 3 h in a sealed bottle, and the methanol (3 mL) was added to remove excess TFA, then the dried hydrolysate was dissolved with ultra-pure water and diluted to 50 mL for analysis. Monosaccharide composition was measured by high performance anion exchange chromatography system (HPAEC, ICS2500, Dionex, Sunnyvale, CA, USA) with a pulsed amperometric detector (PAD), and a Carbopac PA-20 column (3 mm × 150 mm, Dionex, Sunnyvale, CA, USA). The column was eluted with a mobile phase of 2 mm NaOH at a flow rate of 0.45 mL/min under 30 °C, and the injection volume was 25 μL.

### 2.6. Immunostimulatory Activity of Purified Fractions on Inducing NF-κB Activation via Dectin-1 Receptor

HEK-Blue^TM^ hDectin-1b cells were used in this study, and this model system expressed genes of the Dectin-1 and secreted alkaline phosphatase (SEAP) induced by NF-κB [30]. The SEAP activity in the cell supernatant was measured by HEK-Blue^TM^ Detection kit to study sample-related effects on inducing NF-κB activation via the Dectin-1 receptor.

HEK-Blue^TM^ hDectin-1b cells were incubated in DMEM culture media (St. Louis, MO, USA) with 10% fetal calf serum, 50 U/mL penicillin, 100 μg/mL streptomycin and 100 μg/mL normocin (InvivoGen, Toulouse, France) at 37 °C in 5% CO_2_. After logarithmic growth phase of the cell culture, the cells were suspended in a detection culture medium (3 × 10^5^ cells/mL). Then the cells (180 μL) were seeded in 96-well plates and incubated with 20 μL sample solutions (200 μg/mL). Meanwhile, 20 μL PBS and 200 μg/mL SG were added to act as negative and positive controls. After 24 h, the absorbance of the culture supernatant was determined at 630 nm. The experiment was performed three times.

### 2.7. FT-IR Analysis

Samples were mixed with KBr powder for IR determination via a Fourier transform infrared (FT-IR) spectrometer (Thermo Fisher Scientific, Waltham, MA, USA) in the range of 4000–500 cm^−1^.

### 2.8. Methylation Analysis

Methylation analysis of samples was conducted according to the method of previous reports [31]. The partially methylated alditol acetates (PMAA) were analyzed using a GC-MS system (Thermo Finnigan, Bremen, Germany) with HP-5MS silica capillary column (30 m × 250 μm × 0.25 μm), and a temperature program from 180 to 270 °C at 20 °C /min, with a holding stage at 270 °C for 25 min. The molar percentage of each methylated residue was calculated based on the ratios of peak areas.

### 2.9. NMR Analysis

The samples (NSP-30E, KSP-30E, 20E) were dissolved in a mixture of deuterium oxide (D_2_O) and Me_2_SO-*d*6 (1: 6, *v*/*v*) at a concentration of 30 mg/mL to obtain high-resolution signals [32]. The 30E, 40E and 50E were dissolved and freeze-drying performed twice with D_2_O, then dissolved in deuterium oxide for nuclear magnetic resonance (NMR) analysis. ^1^H NMR spectra and ^13^C NMR spectra were recorded using a 600 MHz Varian VNMRS NMR spectrometer (Agilent, Palo Alto, CA, USA) at 70 °C.

### 2.10. Statistical Analysis

Data were expressed as mean ± standard deviation. One-way analysis of variance (ANOVA) was used to analyze differences between groups on a significance level of *p* < 0.05.

## 3. Results and Discussion

### 3.1. Characterization Analysis of Cell Wall Polysaccharides Extracted with Different Methods

#### 3.1.1. Comparison of the Yields, Sugar and β-Glucan Content

The yields, sugar and β-glucan content of cell wall polysaccharides extracted with different methods were listed in Table 1. Alkali extracts contained higher yields (over 8.8%) with sugar content higher than 88%, and the water extract (SP) obtained by superfine grinding contained the highest yield (20.80%) with the sugar content of 78.16%, which might provide the reason that superfine grinding contributed to the release of cell wall polysaccharides [33]; however, it showed a significant difference in the β-glucan content. Among these fractions, the β-glucan content of SP with the value of 19.35% was significantly lower than those of NSP (50.10%) and KSP (87.96%). The previous report showed that superfine grinding could improve the yields of β-glucan from *Ganoderma lucidum* [34], but in our work the β-glucan content of SP obtained using superfine grinding was low, which might be due to differences of fungal cell wall composition.

#### 3.1.2. Molecular Weight Distribution and Monosaccharide Composition Analysis

HPSEC profiles of different extracts are shown in Figure 2, and the weight average molecular weight (*M*_w_), the number average molecular weight (*M*_n_) and polydispersity (*M*_w_/*M*_n_) are listed in Table 2. The results indicated that molecular weight distributions of different extracts were different. The alkali and superfine grinding extracts all showed two peaks with the *M*_w_ of 3.30 × 10^6^~1.44 × 10^7^ g/mol (Peak1) and 5.81 × 10^5^~2.50 × 10^6^ g/mol (Peak2), while the acid extract presented one wide peak with *M*_w_ of 2.59 × 10^5^ g/mol. Moreover, the *M*_w_ of extracts prepared by superfine grinding was higher than other extracts, indicating the superfine grinding treatment can enhance the release of polysaccharides with high molecular weight.

Monosaccharide analysis (Table 3) showed that the polysaccharide fractions prepared by different extraction methods were mainly composed of glucose, followed by galactose. However, the small amount of xylose was found only in acid–alkali extracts, while some mannose was detected only in superfine grinding extracts. These results further confirmed that extraction methods affected the monosaccharide composition of polysaccharides [35].

### 3.2. Structure and Activity Analysis of Ethanol Precipitated Fraction

#### 3.2.1. Comparison of Purified Polysaccharide Fractions from Different Extracts

Ethanol precipitation methods were successfully used to purify polysaccharides from alkaline extracts (NSP and KSP) and superfine-grinding assisted water extract (SP). The sugar contents of purified polysaccharides were both above 82%, especially for KSP-30E and 50E, the sugar content reached 94.28% and 94.26%, respectively. UV wavelength scanning spectra of polysaccharide fractions (Appendix A) showed there were no obvious peaks detected at 280 nm for 30E, 40E and 50E, and extremely weak peaks occurred at around 280nm for KSP-30E, NSP-30E and 20E fractions, indicating that these three fractions might contain little proteins. HPSEC profiles of purified fractions from SP (Figure 3B) showed that two peaks appeared in 20E, while only one signal peak was present in 30E, 40E and 50E with the *M*_w_ ranging from 7.19 × 10^5^ g/mol to 4.16 × 10^6^ g/mol. For polysaccharides purified from alkaline extracts, KSP-30E and NSP-30E showed a similar symmetrical peak in HPSEC profiles (Figure 3A) with the molecular weights of 3.27 × 10^6^ and 2.87 × 10^6^ g/mol, respectively. Moreover, the *M*_w_/*M*_n_ values (1.41–1.64) of purified fractions suggested these fractions had a narrow-ranged distribution of *M*_w_. The relationship of *M*_w_ with radius of gyration (*R_g_*) was established using linear regression, and the slope of *R_g_*-*M*_w_ could represent the chain conformation of polymers. The slope of rigid chain in solvent is higher than 1.0, and the values of 0.33 and 0.5–0.6 represent the conformation of sphere chain and random coil, respectively [36]. The conformation plot slopes of KSP-30E and NSP-30E (Figure 3C) were 1.10 and 1.15, indicating the rigid chain conformation of these polysaccharides [37]. However, the 30E and 40E (Figure 3D) existed as sphere chain conformation with the values of 0.26 and 0.32, and the slope value (0.53) of *R_g_* vs. *M*_w_ suggested a random coil conformation of 50E [38].

The monosaccharides composition results (Figure 4) revealed that all fractions were mainly composed of glucose with a molar percent higher than 80%. Xylose was only detected in NSP-30E, while little fucose was determined only in the fractions of superfine grinding. Moreover, for alkali-extracted fractions, the monosaccharide composition of NSP-30E was more complex than KSP-30E as some xylose and mannose were detected in the former. The main monosaccharide of separated fractions (20E, 30E, 40E, 50E) obtained using superfine grinding was glucose (80.10–98.19%), as well as a small amount of galactose (1.57–16.66%) and fucose (0.25–1.00%). High percentage of glucose indicated that all the purified polysaccharide fractions might be composed of glucans.

#### 3.2.2. Activity Determination on Activating Dectin-1 Receptor

As depicted from Figure 5, all fractions exhibited the activity of activating Dectin-1 receptor in vitro, but there was a significant difference in the strength of activity. The results showed that polysaccharide fractions with similar *M*_w_ exhibited a different activity on activating Dectin-1 reporter, which might be due to the structural difference. Among the purified polysaccharide fractions, KSP-30E showed the strongest activity on activating Dectin-1, which might be related to its structure. Thus, it remains necessary to investigate the structural characteristics of these polysaccharides.

#### 3.2.3. FT-IR Analysis

FT-IR spectra of polysaccharides are shown in Figure 6. The results showed that these samples had the typical characteristic peaks of polysaccharides. The strong absorption band at 3363 and 1241 cm^−1^ represent the stretching vibration and variable angle vibration of O-H, respectively. The peaks at 1092 cm^−1^ corresponded to the C-O-C vibration, and the band at 1641 cm^−1^ showed the stretching vibration of C=O. The absorption band at 2917 cm^−1^ represents the stretching vibration from C-H. Meanwhile, there was no peak at 1610.5 cm^−1^, which represented the N-H variable angle vibration of -CONH-, indicating that these samples were polysaccharides that contained little protein.

#### 3.2.4. Methylation Analysis

Results of methylation analysis are shown in Table 4. The alkali-extracted cell wall polysaccharides consisted mainly of terminal-linked (20.70%–27.86%), (1→3)-linked (40.17%–43.09%) and (1→3, 1→6)-linked glucopyranosyl residues (20.98%–25.45%) with different molar percentage ratios. Besides, small amounts of 1,4-linked Glc*p* and 1,6-linked Gal*p* appeared in NSP-30E, suggesting that the structure of NSP-30E was different from KSP-30E. For the ethanol precipitated fractions obtained from the superfine grinding assisted water extract, excluding the linkage types mentioned above, 20E also consisted of (1→4)-linked (40.15%) and (1→4, 1→6)-linked glucopyranosyl residues (3.79%), indicating that 20E might be a mixture of (1→3)-glucan and (1→4)-glucan. However, 30E, 40E and 50E mainly contained terminal-linked (13.86%–20.86%), (1→4)-linked (62.68%–71.15%) and (1→4, 1→6)-linked glucopyranosyl residues (9.43%–10.10%), which was consistent with the previous results regarding cell wall polysaccharides from *Pleurotus tuber regium* sclerotium [39]. Moreover, it showed that these polysaccharide fractions might possess different primary structural due to different extraction methods and need to be further studied.

#### 3.2.5. NMR Analysis

The 1D ^1^H and ^13^C NMR spectra were used to further elucidate the structure of polysaccharide fractions. For KSP-30E, the chemical shifts of *δ* 4.53 and *δ* 4.51 ppm were found in ^1^H NMR spectrum, indicating the β-configuration for glucopyranosyl residues [40]. Two anomeric proton signals at around *δ* 103 and *δ* 103.09 ppm were observed in ^13^C NMR spectrum (Figure 7G), and the signals shifted to downfield at around *δ* 85 ppm suggested O-substituted C-3, which represented the presence of β-(1→3)-D-Glc*p* [41,42]. NSP-30E showed similar chemical shift signals (*δ* 4.52 and *δ* 103.03 ppm) in ^1^H NMR and ^13^C NMR (Figure 7B,H), but it also displayed one anomeric proton signal at δ 5.07 ppm (Figure 7B) and two chemical shift signals at *δ* 101.06 and *δ* 99.84 ppm in ^13^C NMR (Figure 7H), which indicated that β-configuration and α-configuration appeared in NSP-30E at same time. In addition, the signals of *δ* 85.01 and *δ* 76.68 ppm suggested the existence of (1→3)-D-Glc*p* and (1→4)-D-Glc*p* in NSP-30E (Figure 7H). For polysaccharide fractions prepared by superfine grinding, the signals of β-configuration (*δ* 4.52 and *δ* 103.18 ppm) and α-configuration (*δ* 5.10 and *δ* 100.92 ppm) were detected in 20E (Figure 7C,I), while only signals of α-configuration (*δ* 5.46/5.36/5.35 and *δ* 100.13/99.94/100.23 ppm) were found in 30E, 40E and 50E (Figure 7D–F,J–L). Moreover, the downfield shifted signals at around *δ* 77 ppm represented the presence of (1→4)-linked -D-Glc*p* in these polysaccharides isolated from superfine grinding extraction.

Therefore, considering all the information provided by NMR, methylation analysis and monosaccharides analysis, we speculated that the KSP-30E was β-(1→3)-glucan attached by β- (1→6)-D-Glc*p* as branches, while NSP-30E was mainly composed of β-(1→3)- Glc*p* with β-(1→6)-D-Glc*p* and a few α-linked-Glc*p*. However, the fractions (30E, 40E, 50E) obtained from superfine grinding extraction mainly consisted of α- (1→4)-D-Glc*p* with α- (1→6)-D-Glc*p* as a side chain, except 20E, which mainly contained β-(1→3)-linked-Glc*p* and α-(1→4)-linked-Glc*p*.

Therefore, considering all the information provided by NMR, methylation analysis and monosaccharides analysis, we speculated that the KSP-30E was β-(1→3)-glucan attached by β- (1→6)-D-Glc*p* as branches, while NSP-30E was mainly composed of β-(1→3)- Glc*p* with β-(1→6)-D-Glc*p* and a few α-linked-Glc*p*. However, the fractions (30E, 40E, 50E) obtained from superfine grinding extraction mainly consisted of α- (1→4)-D-Glc*p* with α- (1→6)-D-Glc*p* as a side chain, except 20E, which mainly contained β-(1→3)-linked-Glc*p* and α-(1→4)-linked-Glc*p*.

By comparing the results of activity and structure, we found that β-(1→3)-D-glucan possessed higher immunostimulatory activity than α-(1→4)-glucan, and the activity of α--glucan decreased with a decreasing molecular weight. These results demonstrated that the immunomodulatory activity on activating Dectin-1 receptor of polysaccharides was related to their structures. Thus*,* we further analyzed the structure of β-(1→3)-D-glucan (KSP-30E) with the strongest activity. The chemical shifts of ^1^H and ^13^C NMR from four residues (A, B, C and D) are summarized in Table 5 based on the COSY (Figure 8A), TOCSY (Figure 8B), HMQC (Figure 8C) spectra. According to the signals of chemical shift and relevant literature reports, residues A and B were designated as (1→3)-linked-*β*-D-glucose, while C and D were (1→3,1*→6*)-linked-*β*-D-glucose and terminal linked-*β*-D-glucose, respectively [40]. In addition, the signals of cross-peaks in the HMBC spectrum (Figure 8D) indicated that residues A and B were connected by 1→3-linkage, which was similar to the connection of residue B and C. The signals of DH1-CC6 and DC1-CH6 suggested that the residues C and D were connected via 1→6-linkage. Thus, the repeating unit of KSP-30E was identified as β-(1→3)-glucan with β-(1→6)-linked glucose as branches on every third residue (as shown in Figure 9).

## 4. Conclusions

In this study, different cell wall polysaccharide fractions were extracted and isolated from water extracted residues of *Sparassis latifolia* fruit bodies. The β-glucan content of alkaline extracts was higher than those of acid and superfine grinding assisted water extracts, and KOH extract possessed the highest content of β-glucan with a value of 87.96%. Structural analysis showed that the purified fraction from KOH extract was identified as β-(1→3)-glucan with β-(1→6)-linked glucose as branches, while the purified polysaccharide from NaOH extract was mainly composed of β-(1→3)-glucan with a few α-linkage types. Moreover, superfine grinding-assisted extraction was used to obtain cell wall polysaccharides, and the β-glucan content was notably lower than alkali extract, although it could yield fractions with a high sugar content. Further results showed that superfine grinding-extracted polysaccharides mainly contained α-(1→4)-glucan with little β-(1→3)-glucan. Meanwhile, the *R_g_*-*M*_w_ slope of fractions indicated that the conformation of KSP-30E and NSP-30E existed as a rigid chain. However, the 30E/40E and 50E presented as a sphere chain and a random coil conformation, respectively. Activity analysis of polysaccharides indicated that alkali-extracted fractions displayed stronger immunomodulatory activity by stimulating Dectin-1 receptor in vitro than superfine grinding-extracted fractions, and KSP-30E exhibited the highest immune-enhancing bioactivities, which was elucidated as β-(1→3)-glucan with β-(1→6)-linked glucose as branches attached at every third residue. The findings indicated that the chemical characteristics and bioactivities of cell wall polysaccharides from *Sparassis latifolia* were influenced by different extraction methods. This study provided an effective method for preparation of bioactive β-(1→3)-glucan from the water-extracted residues of *Sparassis latifolia* fruit bodies and would be helpful for its further application in future.

## Figures and Tables

**Figure 1 polymers-14-00549-f001:**
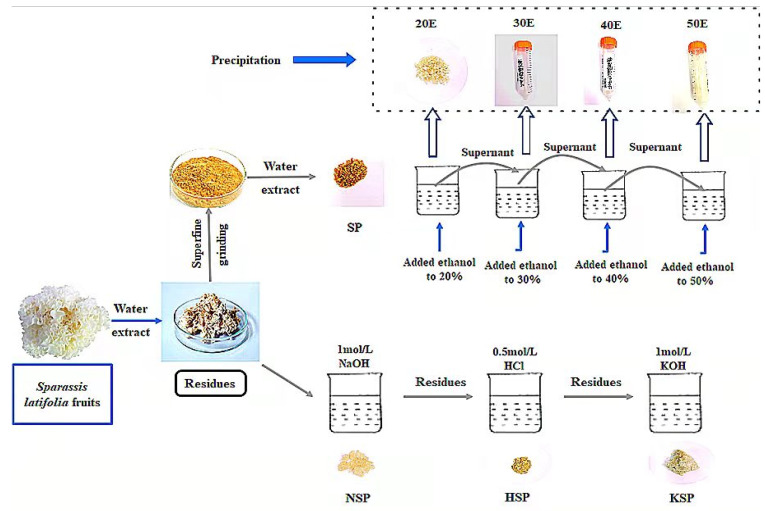
Extraction and separation procedure of cell wall polysaccharides from *Sparassis latifolia* fruit bodies. The polysaccharide fractions of 20E, 30E, 40E and 50E were obtained by sequentially adding ethanol to the different concentrations (20%–50%) in water extract of superfine grinding assisted extraction (SP). NSP: sodium hydroxide extract, KSP: potassium hydroxide extract, HSP: hydrochloric acid extract.

**Figure 2 polymers-14-00549-f002:**
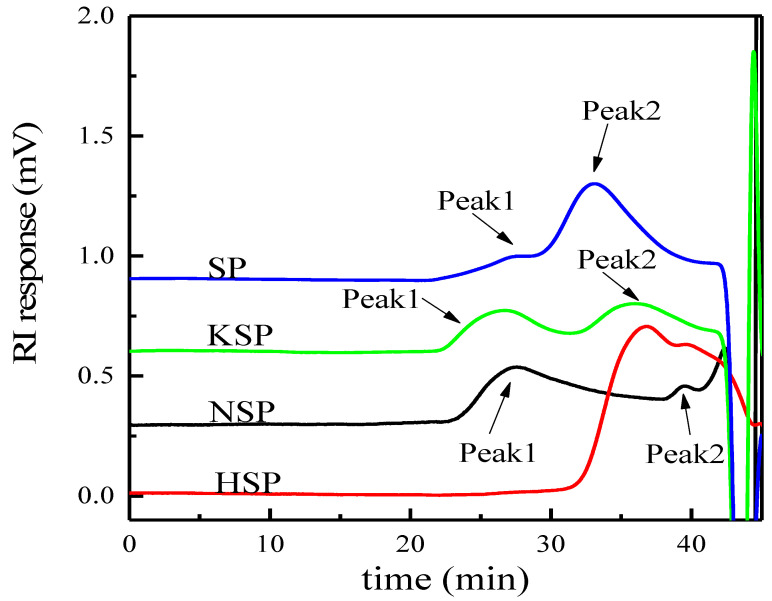
HPSEC profiles of different extracts. The notes of NSP, HSP, KSP and SP are the same as Figure 1.

**Figure 3 polymers-14-00549-f003:**
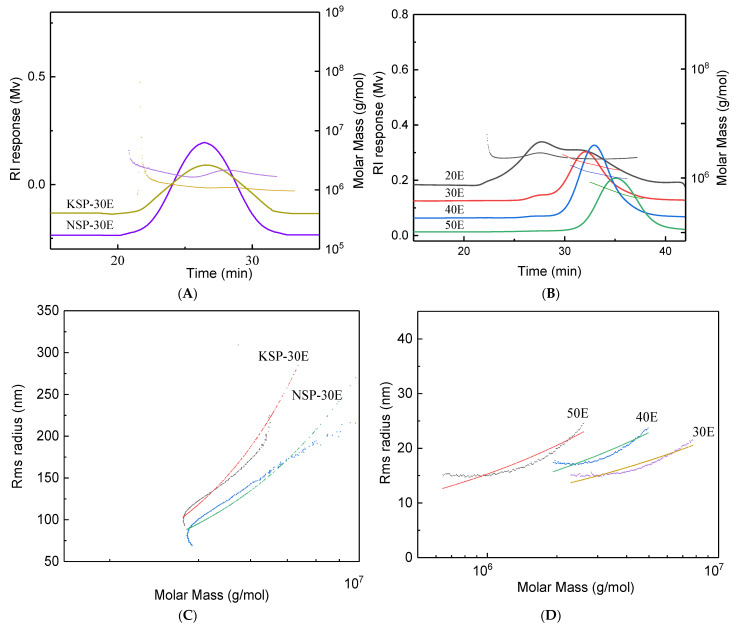
HPSEC profiles, molecular mass distribution and conformational characters of purified fractions. HPSEC profiles and molecular mass of purified fractions isolated from alkali extracts (**A**) and superfine grinding assisted water extract (**B**). The double-logarithmic plot of *R_g_* against *M*_w_ for purified fractions of alkali extracts (**C**) and superfine grinding assisted water extracts (**D**). The fractions of KSP-30E and NSP-30E were obtained by adding ethanol to the final concentration of 30% in KSP and NSP, respectively. The notes of NSP, HSP, KSP,20E, 30E, 40E and 50E are the same as Figure 1.

**Figure 4 polymers-14-00549-f004:**
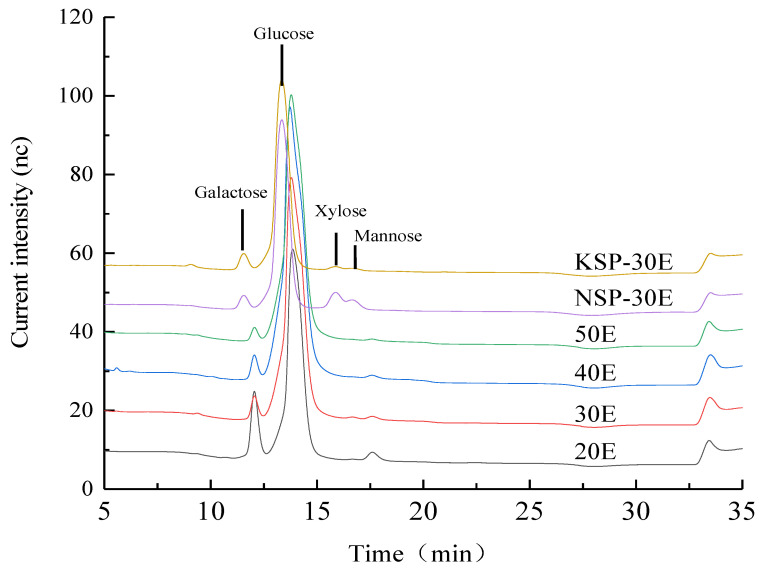
High performance anion exchange chromatography of purified polysaccharides. The notes of NSP-30E, KSP-30E,20E, 30E, 40E and 50E are the same as Figure 3.

**Figure 5 polymers-14-00549-f005:**
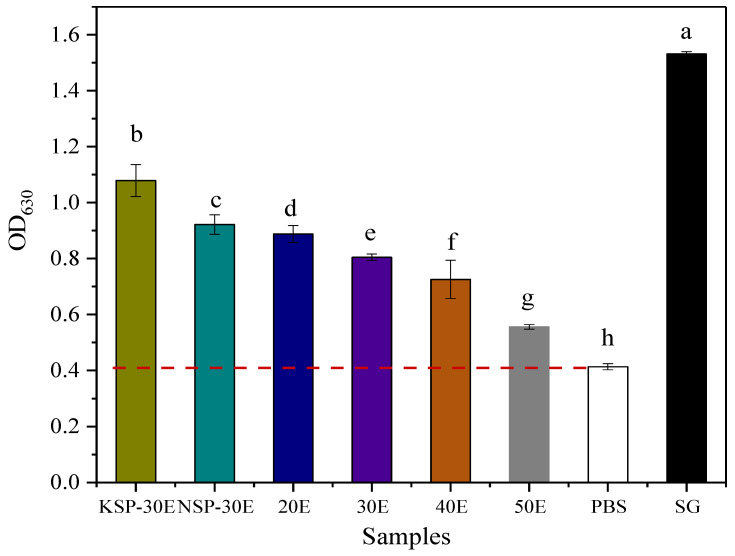
Activity of polysaccharide fractions on activating Dectin-1 receptor; different letters represented significant differences among activities on inducing Dectin-1 of polysaccharides prepared by different extraction methods. OD_630_: The OD value at 630 nm. PBS: Phosphate buffer. SG: Scleroglucan. The notes of NSP-30E, KSP-30E,20E, 30E, 40E and 50E are the same as Figure 3. Different lowercase letters indicate a significant difference at *p* < 0.05.

**Figure 6 polymers-14-00549-f006:**
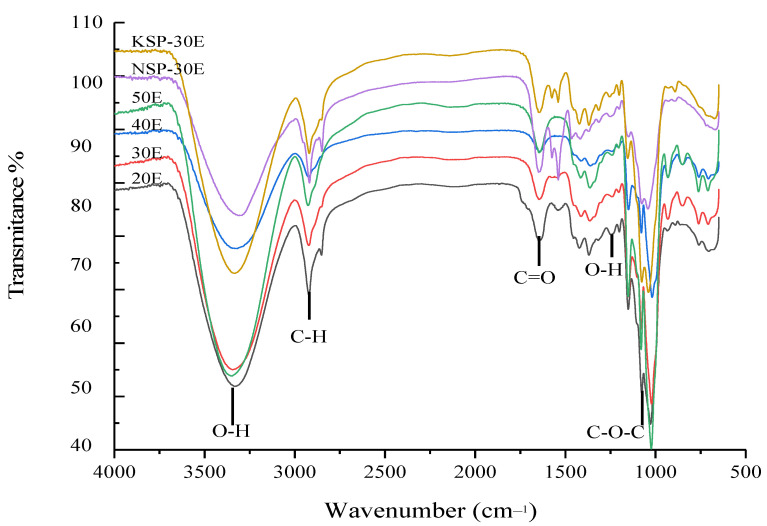
IR spectra of polysaccharide fractions. The notes of NSP-30E, KSP-30E,20E, 30E, 40E and 50E are the same as Figure 3.

**Figure 7 polymers-14-00549-f007:**
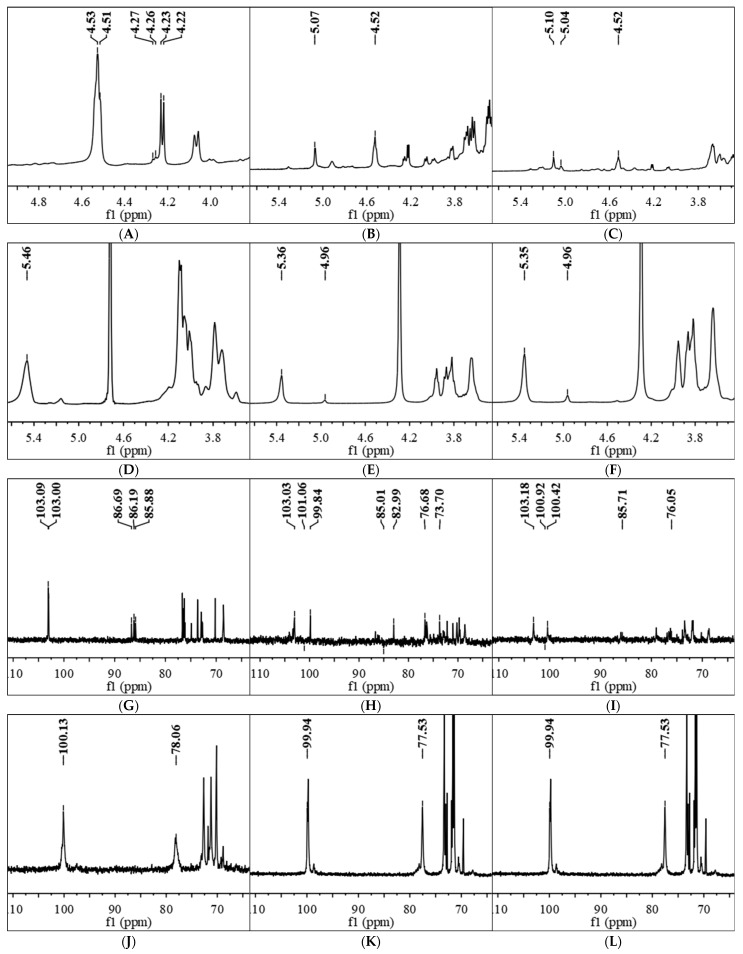
NMR spectra of purified polysaccharides fractions.^1^H NMR spectra of KSP-30E (**A**), NSP-30E (**B**), 20E (**C**), 30E (**D**), 40E (**E**), 50E (**F**) prepared by ethanol precipitation. ^13^C NMR spectra of KSP-30E (**G**), NSP-30E (**H**), 20E (**I**), 30E (**J**), 40E (**K**), 50E (**L**) prepared by ethanol precipitation. The notes of NSP-30E, KSP-30E,20E, 30E, 40E and 50E are the same as Figure 3.

**Figure 8 polymers-14-00549-f008:**
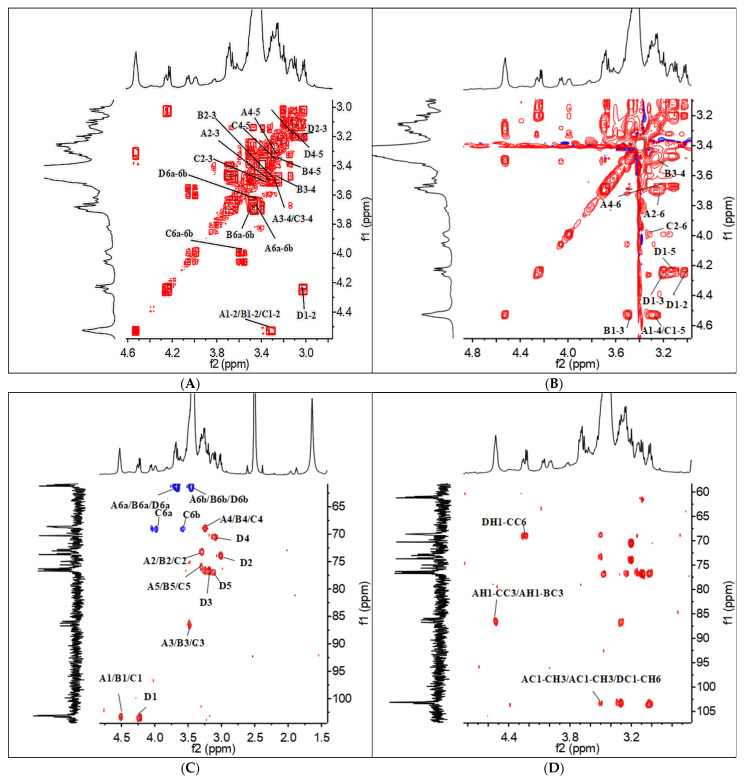
2D NMR spectra of KSP-30E. (**A**–**D**) represent COSY, TOCSY, HSQC and HMBC spectrum, respectively.

**Figure 9 polymers-14-00549-f009:**
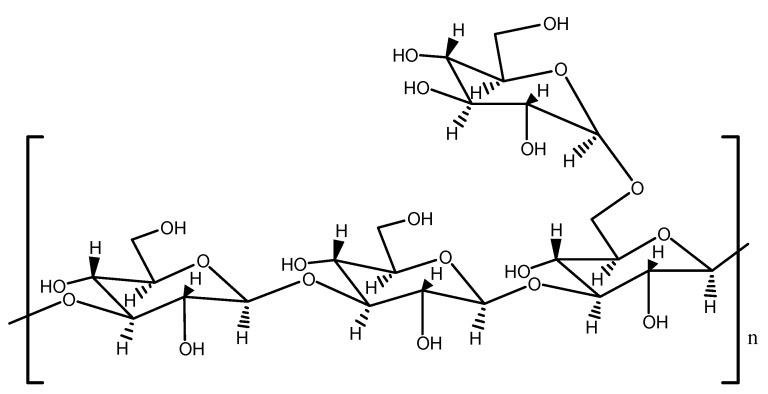
Structure of KSP-30E obtained from *Sparassis latifolia* fruit bodies.

**Table 1 polymers-14-00549-t001:** The yields, sugar and β-glucan content of different fractions.

Extraction Methods	Samples	Yields (%)	Sugar Content (%)	β-Glucan Content (%)
Alkali-acid-alkali	NSP	8.90	88.17 ± 2.09	50.10 ± 0.43
	HSP	2.74	81.37 ± 0.28	16.66 ± 0.29
	KSP	8.83	93.36 ± 1.13	87.96 ± 1.03
Superfine grinding	SP	20.80	78.16 ± 2.15	19.35 ± 1.20

The notes of NSP, HSP, KSP and SP are the same as Figure 1.

**Table 2 polymers-14-00549-t002:** Molecular weight and polydispersity of different extracts.

Extraction Methods	Samples	Peak	*M*_w_ (g/mol)	*M*_n_ (g/mol)	Polydispersity (*M_w_/M_n_*)
Alkali-acid-alkali	NSP	Peak1	3.32 × 10^6^	2.44 × 10^6^	1.36
		Peak2	6.36 × 10^5^	6.15 × 10^5^	1.03
	HSP	Peak	2.59 × 10^5^	4.65 × 10^4^	5.57
	KSP	Peak1	3.30 × 10^6^	2.93 × 10^6^	1.13
		Peak2	5.81 × 10^5^	3.73 × 10^5^	1.56
Superfine grinding	SP	Peak1	1.44 × 10^7^	1.20 × 10^7^	1.20
		Peak2	2.50 × 10^6^	1.54 × 10^6^	1.62

The notes of NSP, HSP, KSP and SP are the same as Figure 1.

**Table 3 polymers-14-00549-t003:** Monosaccharide composition of polysaccharide fractions.

Extraction Methods	Samples	Monosaccharide (mol%)					
		Fucose	Glucosamine	Galactose	Glucose	Xylose	Mannose
Alkali-acid -alkali	NSP	2.56	1.47	15.56	76.35	3.3	-
	HSP	1.01	-	4.38	93.53	0.97	-
	KSP	1.18	-	4.20	91.66	2.95	-
Superfine grinding	SP	0.62	-	4.54	92.58	-	2.25

-: not detected. The notes of NSP, HSP, KSP and SP are the same as Figure 1.

**Table 4 polymers-14-00549-t004:** Methylation analysis of polysaccharide fractions.

Linkage Types (Mol %)	Samples					
	NSP-30E	KSP-30E	20E	30E	40E	50E
Terminal-linked-Glc*p*	27.86	20.70	19.91	20.86	16.12	13.86
1,3-linked Glc*p*	43.09	40.17	17.63	-	-	-
1,3-linked Hex*p*	3.12	-	-	-	-	-
1,4-linked Glc*p*	3.95	-	40.15	69.04	71.15	62.68
1,6-linked Glc*p*	6.19	5.38	8.56	-	-	1.96
1,6-linked Gal*p*	2.24	-	3.67	-	-	-
1,3,4-linked Glc*p*	-	-	2.59	-	3.30	11.69
1,2,3-linked Glc*p*	1.88	6.50	-	-	-	-
1,4,6-linked Glc*p*	-	-	3.79	10.10	9.43	9.81
1,3,6-linked Glc*p*	20.98	25.45	7.38	-	-	-

-: not detected. The notes of NSP-30E, KSP-30E,20E, 30E, 40E and 50E are the same as Figure 3.

**Table 5 polymers-14-00549-t005:** ^1^H NMR and ^13^C NMR chemical shifts of KSP-30E.

Residues	Sugar Linkage	H1/C1	H2/C2	H3/C3	H4/C4	H5/C5	H6a-H6b/C6
A	(1→3)-Linked	4.53	3.30	3.50	3.26	3.32	3.68, 3.45
	-β-Glcp	103.08	73.47	86.18	68.76	76.24	61.30
B	(1→3)-Linked	4.52	3.30	3.47	3.24	3.33	3.7, 3.51
	-β-Glcp	103.19	73.21	86.75	68.52	76.56	61.20
C	(1→3,1→6)-Linked	4.51	3.34	3.50	3.27	3.33	3.97, 3.57
	-β-Glcp	102.93	73.30	86.38	69.50	76.13	69.04
D	Terminal Linked	4.24	3.01	3.19	3.10	3.14	3.64, 3.42
	β-Glcp	103.14	74.15	76.29	70.86	76.87	61.35

## Data Availability

The data presented in this study are available on request from the corresponding author.

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
