# Peer review of "Research on Extraction, Structure Characterization and Immunostimulatory Activity of Cell Wall Polysaccharides from Sparassis latifolia"

_polymers, 2022, doi:10.3390/polym14030549_

Round 1

Reviewer 1 Report

The manuscript Polymers-1454725 entitled "Research on extraction, structure characterization and immunostimulatory activity of cell wall polysaccharides from Sparassis latifolia" after performed the extraction and isolation of several polysaccharide fractions, describes the characterization of these fractions by 1H NMR, considering that alpha- and beta-glucans have been previously described it is convenient to include references for this NMR characterization. The 1H NMR spectra shown in the manuscript have broad signals which are not characteristics of a 600 MHz spectrometer. These broad signals may be due to inadequate shimming or fast relaxation. Authors may consider these differences in broad signals as a consequence of diferences in chemical structures?

Reviewer 2 Report

The current manuscript provides an incomplete and non-confirmatory account of a cell wall polysaccharide extracted from a natural source. I have some major concerns with the study as follows:

1. Various residues of the polysaccharide are duly quantified - however, I didn't find a defined structure of the polysaccharide and should have been provided.

2. The immunostimulatory activities have not been mentioned in the abstract and are in essence incomplete. The research in this area has now advanced to immunomodulation with M1/M2 polarization. The immunofluorescence morphologies of the cultured cells to examine the expression of associated gene markers are missing from the study.

Reviewer 3 Report

The paper raises interesting issues. Until recently polysaccharides, in comparison to e.g. proteins, were very neglected when it comes to biological research and applications. There were a few polymers that were used, such as chitosan or dextran, and the rest of the rich family of polysaccharides was of little interest to the scientific community. This work shows the richness of this group and the possibility of extracting immunologically interesting compounds from fungal fruiting bodies.

 However, I see one problem when it comes to the scientific side of the paper that I think the authors should address before the paper can be published. When isolating macromolecules from fungal material using aqueous solutions, one must always keep in mind contamination with proteins and their degradation products. Such contaminants will have biological activity specifically related to the immune system. The authors do not consider these aspects at all. Additionally, what about the fraction of tested materials that is not soluble in the solvent in which NMR spectra are done. Are you sure all of the extracted polysaccharides for a given fraction are soluble?

Some solid state measurements such as IR or elemental analysis would be useful. Such results and correlations between the results of elemental analysis and the deproteination of the sample obtained from the fungal material can be found by the authors e.g. in the paper:

Pharmaceuticals, 2021, 14(9), 838 “Synthesis and study of antifungal properties of new cationic beta-glucan derivatives”

The work would be worth citing when it comes to this aspect.

Technical Notes

The paper is not technically presented in full accordance with the template. For example, there is no information on conflicts of interest or the contributions of individual authors. This must be completed before publication. Other than that, the paper is written correctly.

Round 2

Reviewer 2 Report

The authors failed to address the comments and concerns raised by the reviewer. In their rebuttal, the authors stated that

"we haven’t elucidated the structure of those polysaccharides in detail due to the limited length of the article"

and 

"Meanwhile, due to the limited space of the article, our study did not investigate the mechanism of immune activity mechanism, and the mechanism of the fraction with better activity"

The space limitation is not a limitation at all in MDPI journals as they are online only journals and have no page limits.

Both the above two points are important for the study and the studies should be conducted given that there is no page limit in Polymers.

Reviewer 3 Report

The authors, in my opinion, have inadequately addressed my concerns. The issue of potential contamination of samples with insoluble compounds especially proteins was overlooked. One method of extract analysis is insufficient. In addition, SEC shows that the samples are heterogeneous in terms of compositions. The authors' comments about the lack of absorbance at 280nm were not supported by the results although such results would only apply to soluble proteins anyway.Before publication, the paper should be completed with the data I requested.

Round 3

Reviewer 2 Report

No further comments.

Reviewer 3 Report

I am glad that the authors took my comments into consideration and interacted with my argument . In my opinion, the work has benefited and can be published in this form.